# Effect of Chloroplast ATP Synthase on Reactive Oxygen Species Metabolism in Cotton

**DOI:** 10.3390/ijms252312707

**Published:** 2024-11-26

**Authors:** Li Zhang, Panpan Jing, Biao Geng, Jinlong Zhang, Jinjiang Shi, Dong Liang, Yujie Yang, Yunfang Qu, Jinling Huang

**Affiliations:** College of Agriculture, Shanxi Agricultural University, Taigu, Jinzhong 030801, China; 13935238906@163.com (L.Z.); ksxj2022@163.com (P.J.); 13263246871@163.com (B.G.); 18434304383@163.com (J.Z.); 18678021516@163.com (J.S.); liangdong2262@163.com (D.L.); 15039490151@163.com (Y.Y.); quyunfang@163.com (Y.Q.)

**Keywords:** cytoplasmic male sterility, programmed cell death, reactive oxygen species, ATP synthase, chloroplast genome

## Abstract

Abnormal programmed cell death in the tapetum is induced by reactive oxygen species (ROS), which are the main factors leading to cytoplasmic male sterility (CMS). These abnormalities are caused by genetic interactions between nuclear and cytoplasmic genes. To explore the role of chloroplast genes in ROS metabolism, next-generation and single-molecule real-time sequencing of the chloroplast genome were performed in the cotton CMS line Jin A (Jin A-CMS). Our results showed that the chloroplast genome is 160,042 bp in length and consists of 131 genes, including 112 functional genes. An analysis of the functional annotation and sequence comparison with the *Gossypium hirsutum* chloroplast genome as a reference revealed that 29 genes in Jin A-CMS have single-nucleotide polymorphisms, including subunits of ATP synthase, NAD(P)H-quinone redox reductase, and photosystem complexes. Compared to the Jin B maintainer, the anthers of Jin A-CMS at the microspore abortion stage have significantly lower expression of *atpB*, *atpE*, and *atpF*. The relative expression of these genes is significantly higher in the three-line F1 hybrids compared to Jin A-CMS. The ROS levels in the leaves increased in response to the silencing of *atpE* and *atpF* in cotton plants. In summary, the results of our study show that the ATP synthase subunit genes *atpE* and *atpF* are closely linked with ROS metabolism. These results provide basic information for the functional analysis of ATP synthase in cotton.

## 1. Introduction

Cytoplasmic male sterility (CMS) is a widespread natural phenomenon in higher plants characterized by maternal inheritance, pollen abortion, and a functional pistil [1,2]. Anther development is a complex process that includes the proliferation and differentiation of the pollen sac multilayer membrane, specific cell apoptosis, microspore mother cell meiosis, microspore proliferation, and development. Higher plants produce specific physiological reactions that result in the death of cells, tissues, or organs in specific parts under changes in the external environment. This process of death induced by external signals leads to the autonomous control of cells, referred to as programmed cell death (PCD). Abnormal PCD in the tapetum during microspore development is the main factor leading to CMS [3,4]. In plants, reactive oxygen species (ROS) are the main inducers of PCD [5]. The excess accumulation of ROS leads to oxidative stress, which triggers the PCD pathway and ultimately leads to cell death.

CMS is a phenomenon caused by genetic interactions between nuclear and cytoplasmic genes [6]. The nuclear background of CMS and its maintainer is homologous, although there are differences in the cytoplasmic genes. The results of previous studies have shown that CMS lines result from mitochondrial gene rearrangements [7,8]. The chloroplast genome of plants is highly conserved in terms of structure, gene number, and gene composition compared with the mitochondrial and nuclear genomes. Despite this, minor changes, such as changes in size, contraction, expansion of repeating regions, and structural rearrangement, still exist [9]. Chloroplast gene expression is involved in plant responses to environmental stresses and is crucial for plant development [10]. The differences in physiological and biochemical indices, chloroplast ultrastructure, relative expression, and genome indicate that chloroplasts may be associated with CMS among rice heterokaryon CMS strains [11].

There is increasing evidence showing that chloroplast ROS are widely involved in responses to various biological and abiotic stresses in plants. Chloroplasts maintain a redox state and regulate ROS metabolism through photosynthesis [12,13]. During photosynthesis, ROS are rapidly produced in chloroplasts, which are the main sites of ROS production in plants [14]. For example, excited electrons in PSI are transferred to oxygen molecules when there is excessive energy and insufficient NADP^+^, an electron acceptor, and superoxide anions (O_2_^−•^) are produced via reduction. More chemically stable hydrogen peroxide (H_2_O_2_) is generated from O_2_^−•^, either spontaneously or with the aid of superoxide dismutase on the stromal side of the thylakoid membrane [15]. When the plastoquinone pool in PSII is in a highly reduced state (e.g., in response to high light, drought, or low CO_2_ concentration), a non-radical form of highly active singlet oxygen (^1^O_2_) is produced [16,17]. During PSII complex repair and reassembly, ^1^O_2_ is generated via the interaction of chlorophyll molecules and the PSII complex [18]. Chloroplast proteins participate in ROS metabolism in CMS lines. The results of several studies have shown that there are significant differences in the ultrastructure [19], DNA levels [20], and protein levels between cytoplasmic male sterile and maintainer lines.

Chloroplasts are essential organelles in higher plants and play important roles in photosynthesis, the metabolism of fatty acids and nitrogen, and internal redox signal transfer [21,22,23]. The chloroplast genome is a typical double-linked ring structure consisting of a small single-copy region and a large single-copy (LSC) region. These two regions are separated by a pair of reverse repeating regions (IRa and IRb) [21]. There are 110–130 genes in the chloroplast, including genes responsible for photosynthesis, self-reproduction, and chloroplast transcription and expression, in addition to some genes of unknown function [24,25]. The study of the chloroplast genome contributes to elucidating the interactions between cytoplasmic genes and physiological and biochemical metabolism at the molecular level.

The main chloroplast thylakoid membrane complexes are ATP synthase complexes composed of CFo and CF1, photosystem I (PSI) complexes, photosystem II (PSII) complexes, and cytochrome 6 complexes. Chloroplast ATP synthase produces ATP via the electrochemical proton gradient generated by photosynthesis. Thereafter, protons pass through the membrane-embedded Fo motor, driving ATP synthesis in the F1 head by rotary catalysis [26]. The nine subunits of chloroplast ATP synthase are encoded by both the chloroplast and nuclear genomes. The α (*atpA*), β (*atpB*), and ε (*atpE*) subunits of CF1 and the six I (*atpF*), III (*atpH*), and IV (*atpI*) subunits of CFo are encoded by chloroplast genes. The γ (*atpC*) and δ (*atpD*) subunits of CF1 and the II (*atpG*) subunit of CFo are encoded by nuclear genes [27]. The six subunits encoded by chloroplast genes are located on two gene clusters, or operons, and the subunit genes are simultaneously transcribed in each cluster. The chloroplast ATP synthase ε subunit is necessary for the recombination of the FoF1 complex. The N-terminal structure of the ε subunit regulates its ability to block transmembrane proton leakage, thereby influencing ATP synthesis [28]. The soluble and bound forms of the ε subunit are potential inhibitors of ATPase and are necessary for maintaining the proton gradient [29,30]. ε is the smallest subunit of chloroplast ATP synthase and is critical for the binding of F1 and Fo and normal H (+) translocation [31]. The combination of the I subunit from CFo with CF1 results in proton transport. Interactions between the δ subunit and the β, γ, ε, I, II, III, and IV subunits of CFo are responsible for preventing proton leakage [32]. The cross-linking of I and II-δ in chloroplast ATP synthase inhibits photophosphorylation and the loss of ATP hydrolytic activity [26]. The ATP synthase genes respond to biotic and abiotic stress in *Arabidopsis* [33,34,35,36].

The main reason for abortion in Jin A-CMS is premature PCD of the anther tapetal layer, which is related to excessive ROS accumulation [5]. In that study, the ATP content decreased significantly at the microspore abortion stage in Jin A-CMS [5]. The results of transcriptomic and proteomic studies have shown differential expression of chloroplast enzymes and genes related to ROS clearance at the key stage of microspore abortion in Jin A-CMS compared to the maintainer Jin B [37,38]. Chloroplast genes and proteins are linked with ROS metabolism in CMS. However, the mechanism by which chloroplast ATP synthase participates in ROS metabolism is largely unknown. To better understand the relationship between ROS metabolism and chloroplast genes, we sequenced the chloroplast genome of Jin A-CMS. The functions of subunit I of chloroplast ATP synthase CFo and several small subunits (β and ε subunits) of CF1 were analyzed to determine the role of chloroplast ATP synthase subunit genes in ROS metabolism. The results of this study provide a basis for further research on chloroplast ATP synthase function.

## 2. Results

### 2.1. Sequencing Data Quality Control and Statistics

The original data volume of the next-generation sequencing (NGS) was 9037.3 Mb, and the effective data volume was 9025 Mb following large-scale trimming of the original data. The percentage of bases with a Phred value greater than 20 was 99.26% of the total bases. The percentage of bases with a Phred value greater than 30 was 97.17%. The GC content was 36.35%.

The number of subreads after filtration was 685,714 in the Jin A SMRT sequencing, the size of the subread data was 967,209,861 bp, and the largest subread length was 38,029 bp. The subread length N50 was 1327 bp. The N90 length of the subreads was 983 bp, whereas the average length of the sample reads was 1411 bp. These results indicated that the constructed database and sequence were suitable for subsequent chloroplast genome assembly and bioinformatics analysis.

### 2.2. Assembly and Characteristics of the Jin A-CMS Chloroplast Genome

Chloroplast DNA is a double-stranded covalent closed-ring molecule in higher plants that varies in length in different species. Following genome assembly, the chloroplast genome length in Jin A-CMS was found to be 160,042 bp (Figure 1). The genome consists of 131 genes, including 112 functional genes (79 protein-coding genes, 29 tRNA genes, and 4 rRNA genes) and 19 repeat genes (Table 1).

The base composition and gene distribution of each component region (LSC/SSC/IR) were determined, and this information is summarized in Table 2. Four typical regions were identified: LSC (55.37%), SSC (12.63%), and two IRs (15.99%).

Functional analysis of the genome revealed that most of the genes are related to photosystem and ATP synthesis (Table 3). There are 5 genes encoding PSI subunits, 15 genes encoding PSII subunits, 12 genes encoding NADH dehydrogenase, 6 genes encoding cytochrome b/f, 6 genes encoding ATP synthase, and 1 gene encoding Rubisco large subunits. In addition, there are 9 genes encoding ribosome large subunit proteins, 12 genes encoding ribosome small subunit proteins, 4 genes encoding DNA-dependent RNA polymerase, 4 genes encoding ribosomal RNAs, and 28 genes encoding transfer RNAs. Other identified genes include a maturase enzyme-encoding gene (*matK*), a protease gene (*clpP1*), an envelope protein-encoding gene *(cemA*), an acetyl-CoA carboxylase gene (*accD*), and a cytochrome synthesis gene (*ccsA*). In addition, five genes of unknown function were identified. Five reference databases were used for gene annotation, among which the Swiss-Prot database was used to annotate the largest number of genes (Table 4), and a total of 86 protein-coding genes (including 79 genes and 7 duplicate genes, all encoding proteins) were annotated.

### 2.3. Comparative Analysis of the Chloroplast Genome in Cotton

The cytoplasmic background of Jin A-CMS is *Gossypium hirsutum*; as such, we chose the chloroplast genome of *Gossypium hirsutum* as a reference [39]. A total of 29 chloroplast genes with single-nucleotide (SNP) differences were obtained through sequence comparison of the chloroplast-coding protein genes between Jin A-CMS and the reference *Gossypium hirsutum* chloroplast genome sequence (Table 5), mainly subunits of ATP synthase, NAD(P)H-quinone oxidoreductase, and photosystem complexes. The results for the comparison of the amino acid sequences of the proteins encoded by these genes are shown in Table 5.

### 2.4. The Relative Expression of ATP Synthase Subunit Genes During Anther Development in Jin A-CMS

ATP synthase plays an important role in cellular energy conversion and transfer. In one study, there was decreased ATP content in the anthers of Jin A-CMS at the microspore abortion stage [5]. To explore the role of chloroplast ATP synthase subunits in Jin A-CMS, we measured the relative expression of three ATP synthase subunit genes, *atpB*, *atpE*, and *atpF*, in Jin A-CMS, the maintainer line Jin B, and the F1 three-line hybrids (Figure 2). The results showed that the expression levels of *atpB*, *atpE,* and *atpF* were significantly lower in the sterile Jin A line than in the maintainer line at the microspore abortion stage. Moreover, the expression levels of these genes were significantly greater in the F1 three-line hybrids than in Jin A-CMS. Therefore, we hypothesized that the differences in the transcription levels of these genes led to the inhibition of ATP synthesis.

### 2.5. ROS Detection in atpB-, atpE-, and atpF-Silenced Cotton Plants

Our results showed that there was significantly increased accumulation of O_2_^−•^ in the leaves of the *atpE-* and *atpF*-silenced plants compared to the negative control plants (Figure 3a,b). There were no significant differences in H_2_O_2_ between the experimental group and the control group (Figure 3c), with similar results for the assay of O_2_^−•^ and H_2_O_2_ contents (Appendix A).

The results of ^1^O_2_ determination in the leaves showed that there was more intense fluorescence in plants silenced by *atpE* and *atpF* than in the negative control, indicating significant accumulation of ^1^O_2_ (Figure 4). There was no difference in ROS content between the *atpB-*silenced cotton plants and the control plants, with similar results for the assays of ^1^O_2_ content (Appendix A).

## 3. Discussion

At present, the complete chloroplast genomes of many important crop species have been sequenced. Phylogenetic analysis of chloroplast genomes in ten species of *Gossypium* showed that the chloroplast genome is relatively conserved, ranging from 159,035 to 160,317 bp in size and composed of four typical regions: LSC, SSC, and two IRs. The variation in IRs is the greatest among different cotton species and is the main determinant of chloroplast genome length differences in cotton [40]. Site-specific selection analysis revealed that certain coding sites in 10 chloroplast genes (*atpB*, *atpE*, *rps2*, *rps3*, *petB*, *petD*, *ccsA*, *cemA*, *ycf1*, and *rbcL*) have undergone evolution in terms of their protein sequence [41]. The chloroplast genome sequence of cotton is similar to that of tobacco, with no rearrangements, AT-rich (with an average content of 62.76%), and showing codon bias toward A and T, especially in the third codon position, with usage as high as 69.31% [40]. In this study, the size of the chloroplast genome of cotton CMS Jin A was characterized as 160,042, with 37.31% GC content. The genome consists of 131 genes, including 112 functional genes (79 protein-coding genes, 29 tRNA genes, and 4 rRNA genes) and 19 repeat genes. These results indicate the high conservation of chloroplast genome evolution in *Gossypium*.

The results of some studies show that chloroplast proteins are involved in physiological and biochemical metabolism in CMS, with significant differences found in ultrastructure [19], DNA levels [20], and protein levels between the CMS and maintainer lines. Chloroplasts are associated with CMS, and there are differences in physiological and biochemical indices, chloroplast ultrastructure, relative expression, and the chloroplast genome among rice heterokaryon CMS strains [11]. We sequenced the chloroplast genome of Jin A-CMS, and comparison with the *Gossypium hirsutum* reference sequence revealed 29 genes with SNPs, which are primarily involved in energy metabolism and photosystem composition.

Chloroplasts, as important organelles for organic matter production in plants, play important roles in plant growth, development, and stress regulation. When the ^1^O_2_ levels in chloroplasts increase, PCD is triggered [42]. Chloroplast genes play a crucial role in the induction of PCD in plants. In transgenic tobacco plants with plastid *ndhF* gene defects, reduced ROS levels are associated with delayed senescence [43]. Loss of function of the gene for FZ1, encoding a membrane GTPase, triggers ROS accumulation via chloroplast membrane damage and is sufficient to initiate the HR signaling cascade in *Arabidopsis* mutants [44].

The ε subunit of chloroplast ATP synthase affects the morphology and structure of the thylakoid membrane near photosystem II through its specific interaction with CF1, which hinders proton loss from the thylakoid membrane [45]. A comparison between the chloroplast genomes of the sterile line Jin A and *Gossypium hirsutum* revealed differences in the nucleotide sequences of the *atpB*, *atpE*, and *atpF* genes (Table 5). The relative expression of genes was measured in microspore buds during microspore abortion. Quantitative qRT-PCR data showed that *atpB*, *atpE*, and *atpF* were significantly downregulated at the microspore abortion stage. H_2_O_2_ and ^1^O_2_ accumulated in the leaves of the *atpE* and *atpF* gene-silenced cotton. The results of a previous study showed that the ATP content of Jin A-CMS anthers was significantly lower than that of maintainer Jin B anthers, and the energy metabolism of Jin A-CMS anthers was disrupted at the microspore abortion stage [5]. We speculated that the ATP synthase genes *atpE* and *atpF* regulate energy metabolism through changes at the transcriptional level in Jin A-CMS. Conversely, the relative expression levels of *atpE* and *atpF* were downregulated and influenced the structure and proton transfer of the photosystem II thylakoid, resulting in ROS accumulation and microspore development. The above findings are consistent with those of Shi [46].

Polymorphism in the chloroplast ATP synthase subunits were associated with a maternally inherited enhanced stress recovery [33,34,35,36,47]. The decrease in the expression levels of ATP synthase genes led to the inhibition of ATP synthesis in Jin A-CMS. This may be caused by decreased levels of oxidative phosphorylation with ADP, uncoupled with oxygen, resulting in ROS accumulation. The lack of energy impeded the chloroplast-related reactions, which led to ROS accumulation. Additional functions of chloroplast genes in plant growth and development await identification.

## 4. Materials and Methods

### 4.1. Plant Materials

In this study, the plant materials were planted in the experimental field of Shanxi Agricultural University from 2020 to 2023. Plants of Jin A-CMS, the homologous heterogenic maintainer Jin B, and the three-line hybrid F1 were managed according to the conventional field management practices for cotton. The cotton plants (Jin B) used for gene silencing were cultured in an artificial climate room (16 h light: 8 h darkness, 8000 lx, 22–23 °C, and 70% relative humidity).

At the flowering stage, buds were collected from Jin A-CMS, maintainer line Jin B, and three-line F1 hybrid cotton plants and classified according to different microspore development stages [48]. Young leaves were taken for use as experimental materials. Thereafter, these materials were quickly frozen in liquid nitrogen and stored at −80 °C.

### 4.2. DNA Sequencing, Data Quality Control, and Statistics

An Illumina TruSeq™ Nano DNA Sample Prep Kit (Illumina, San Diego, CA, USA) was used for library construction. The experimental steps were as follows: total DNA was extracted, with 1 μg used as the starting point for library construction. The Covaris M220 ultrasound was used to shear the DNA into 300–500 bp fragments, with an A-tail added to the 3’ end, and an index adapter was ligated using the TruSeq™ Nano DNA Sample Prep Kit. Library enrichment and PCR amplification were performed for 8 cycles. The purpose strips were recycled using 2% agarose gel (Certified Low Range Ultra Agarose, Bio-Rad, Hercules, CA, USA), and TBS380 Picogreen (Invitrogen, Carlsbad, CA, USA) was used for quantification, according to the proportion of data mixed on the machine. Bridge PCR was performed on a cBot solid phase vector to generate clusters, and an Illumina NovaSeq sequencing platform (Illumina, San Diego, CA, USA) was used for 2 × 150 bp sequencing.

For quality control and statistics of the NGS data, Illumina NovaSeq 6000 sequencing technology was used for paired-end sequencing of the DNA samples. The original sequencing data were processed as follows: polymerase reads lower than 200 bp in length and polymerase reads with masses lower than 0.80 were removed; subreads extracted from polymerase reads and adapter sequences were removed; and subreads lower than 200 bp in length were removed. The sequencing quality value (Phred) was above 30, indicating that the sequencing quality was better (Appendix B). When the library was built more evenly, the fluctuation of the dividing line between the four colors representing the different bases was very little, almost in a straight line (Appendix C). Therefore, the quality of sequencing data is functional, and the error rate is low, which can be used for following experiments.

### 4.3. Chloroplast Genome Assembly and Annotation

First, Illumina sequencing data were assembled using GetOrganelle v1.7.5 software (https://github.com/Kinggerm/GetOrganelle., accessed on 7 November 2022). Next, BWA v0.7.17 was used to align the NGS assembly with PacBio SMRT sequencing data to extract SMRT sequencing data of the target sample, which were mixed with the NGS data for assembly using SPAdes v3.14.1 software. Sequences with sufficient covering depth and long assembly length were selected as candidate sequences, the chloroplast scaffold sequences were confirmed by comparison with the NT library, and the sequences were connected according to overlap.

Next, the clean reads were compared to the chloroplast genome sequence of *Gossypium hirsutum*, and Pilon v1.23 was used to correct the bases. Lastly, the starting position and direction of the chloroplast assembly sequence were determined according to the reference genome, and the 4 chloroplast regions were identified. The final chloroplast genome sequence was obtained by including the large single-copy, small single-copy, and two inverted repeat regions.

GeSeq (https://chlorobox.mpimp-golm.mpg.de/geseq.html/, accessed on 8 November 2022) software was used to predict protein coding and tRNA and rRNA genes of the chloroplast genome (protein search identity: 60; rRNA, tRNA, DNA search identity: 35; 3rd Party tRNA annotators: tRNAscan-SE). The position of each coding gene was determined using BLAST searches against ref cp genes. Manual corrections of genes for start/stop codons and for intron/exon boundaries were performed in SnapGene Viewer by referencing the ref cp genome. The circular chloroplast genome map of JinA-CMS was drawn using OrganellarGenomeDRAW (https://chlorobox.mpimp-golm.mpg.de/OGDraw.html, accessed on 8 November 2022). The preference value of the codon was obtained by calculating the relative synonymous codon usage (RSCU) using Cusp software (EMBOSS v6.6.0.0).

Functional annotations were performed using sequence-similarity BLAST searches with a typical cut-off E-value of 10^−5^ against several publicly available protein databases, namely the NR (http://www.ncbi.nlm.nih.gov/, accessed on 9 November 2022), GO (http://geneontology.org/, accessed on 9 November 2022), eggNOG (http://eggnogdb.embl.de/, accessed on 9 November 2022), KEGG (http://www.genome.jp/kegg/, accessed on 9 November 2022), and Swiss-Prot (https://web.expasy.org/docs/swiss-prot_guideline.html, accessed on 9 November 2022) databases.

### 4.4. Gene Expression

RNA was extracted from the anthers of the three types of cotton materials using a Plant RNA Extraction Kit (Aidlab Biotechnologies Co. Ltd., Beijing, China) according to the manufacturer’s instructions. Anthers (0.1 g) were ground in liquid nitrogen and centrifuged at room temperature, after which absolute ethanol was added to the supernatant for RNA precipitation. After adsorption to the column, protein and pigment impurities were removed. Three distinct bands with concentrations between roughly 600 and 1200 ng/μL were detected via 1% agarose gel electrophoresis, and the A260/A280 ratio was approximately 2.0; thus, the extracted RNA could be used for subsequent experiments. A PrimeScript™ RT Reagent Kit with gDNA Eraser (Perfect Real Time) (Takara Biotech Co., Ltd., Dalian, China) was used for reverse transcription. All regions of RNA can be uniformly synthesized when using an RT Primer Mix combining Random 6-mers and Oligo dT Primer as the primers for reverse transcription.

The relative expression of genes was measured with TB Green^®^Premix Ex Taq™ II (Tli RNaseH Plus) (Takara Biotech Co., Ltd., Dalian, China) using the Bio-Rad CFX Connect™ fluorescence quantitative PCR assay system (Bio-Rad, Hercules, CA, USA). The primer sequences are shown in Appendix D.

### 4.5. Gene Cloning and Carrier Construction

Specific primers were designed using NCBI Primer-BLAST (https://www.ncbi.nlm.nih.gov/tools/primer-blast, accessed on 5 December 2022) (Appendix D). The coding sequence (CDS) of the genes was amplified, and the plasmid pMD19T (Takara Biotech Co., Ltd., Dalian, China) was connected to the CDS of genes for transforming Trans1-T1 cells (TransGen Biotech Co., Ltd., Beijing, China) according to the manufacturer’s instructions. After PCR detection, the positive bacterial solution was sent to a company (Beijing Tsingke Biotech Co., Ltd., China) for sequencing.

The pTRV2 plasmid was linearized using restriction enzymes (New England Biolabs, Ipswich, MA, England) according to the manufacturer’s instructions (primer and restriction enzyme sites in Appendix D). Recombinant vectors were constructed using a ClonExpress II One Step Cloning Kit (Vazyme Biotech Co., Ltd., Nanjing, China).

### 4.6. Virus-Induced Gene Silencing in Cotton

To explore the functions of *atpB*, *atpE,* and *atpF*, recombinant vectors for gene silencing were constructed. An empty pTRV2 vector was used for negative control plants. After 15 days of silencing, the cotton leaves were stained with nitrotetrazolium blue chloride (NBT) and 3,3’-diaminobenzidine (DAB), and the ROS content of the silenced plants was determined (Figure 3a–c). The recombinant vectors for gene silencing were then transformed into *Agrobacterium tumefaciens* GV3101and subsequently injected into cotton cotyledons [49].

### 4.7. ROS Detection

NBT and DAB staining was performed according to a published method [50]. The NBT staining solution was dissolved in double-distilled water at a concentration of 0.2 mM. The samples were soaked in the solution for 2 h and then decolorized using 95% ethanol. The DAB staining solution was dissolved in redistilled water at a concentration of 1 mg/mL, and the pH was adjusted to 3.8 with concentrated hydrochloric acid. The samples were soaked in this solution for 24 h and then decolorized with acetic acid, glycerin, and 95% ethanol (1:1:3). The samples were observed using a stereomicroscope (Olympus, Hamburg, Germany). ^1^O_2_ was measured using Singlet Oxygen Sensor Green reagent (Biolab Technology Co., Ltd., Beijing, China) according to the manufacturer’s instructions. SOSG (10 μM) was used for staining the samples. The samples were soaked in solution for 12 h and observed under a confocal microscope. The excitation and emission wavelengths were 504 nm and 525 nm, respectively.

The determination of H_2_O_2_ and O_2_^−•^ contents was performed as previously described [5]. ^1^O_2_ determination was performed by measuring the dimethyl-4-nitrosoaniline reduction in peak absorption under 440 nm (Plant Singlet Oxygen assay kit, GENMED, Plymouth, MN, USA). Each experiment was independently repeated three times.

### 4.8. Statistical Analysis

All the assays were independently repeated at least three times. Statistical analysis was performed using Student’s *t*-tests. The charts were drawn using GraphPad Prism version 8.0.0 for Windows, GraphPad Software, San Diego, CA, USA.

## 5. Conclusions

The chloroplast genome of Jin A-CMS was 160,042 bp in length and consisted of 131 genes, including 112 functional genes (79 protein-coding genes, 29 tRNA and 4 rRNA genes) and 19 repeat genes. Compared with the proteins encoding chloroplast genes in *Gossypium hirsutum*, there were 29 genes containing SNPs in Jin A-CMS. The relative expression levels of *atpB*, *atpE*, and *atpF* were significantly downregulated in anthers at the microspore abortion stage, and *atpE* and *atpF* gene silencing led to the accumulation of ROS in cotton leaves.

## Figures and Tables

**Figure 1 ijms-25-12707-f001:**
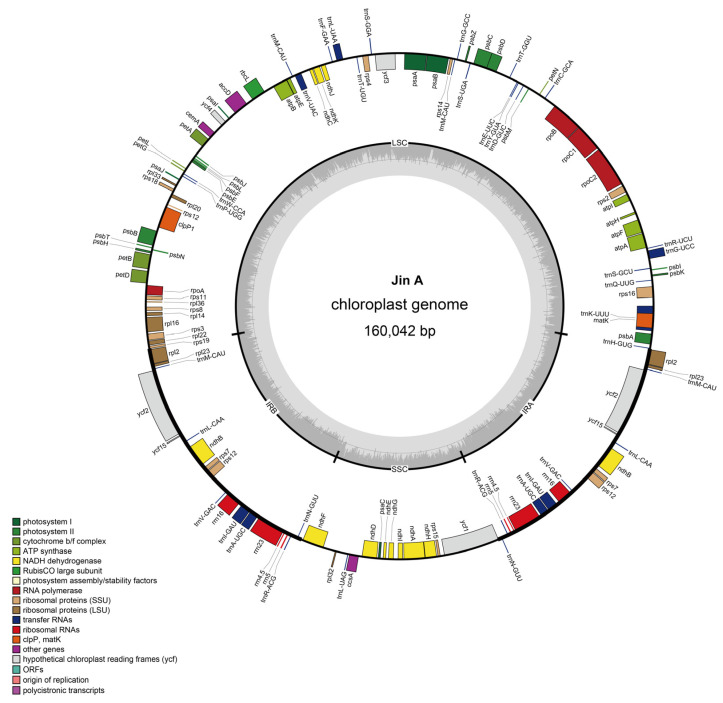
Circular map diagram of the chloroplast genome in Jin A-CMS. The details of the DNA strands transcribed clockwise (+) and counterclockwise (−) are displayed on the inside and outside of the circle, respectively. The color indicates the function of the gene, as shown in the legend at the bottom left of the figure.

**Figure 2 ijms-25-12707-f002:**
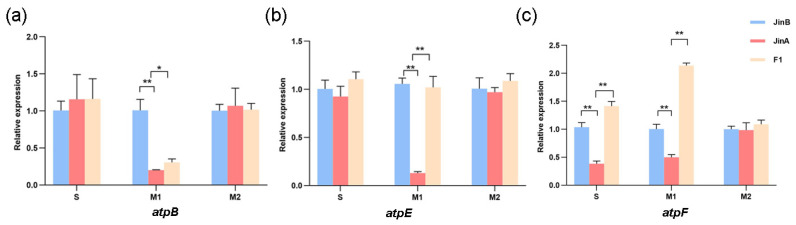
Expression analysis of ATP synthase subunit genes during anther development in Jin A-CMS. (**a**) The relative expression of *atpB*; (**b**) The relative expression of *atpE*; (**c**) The relative expression of *atpF*. S (before microspore abortion stage), M1 (microspore abortion stage) and M2 (after microspore abortion stage). Values are means ± SD of three replicates (* *p <* 0.05; ** *p <* 0.01, Student’s *t*-tests).

**Figure 3 ijms-25-12707-f003:**
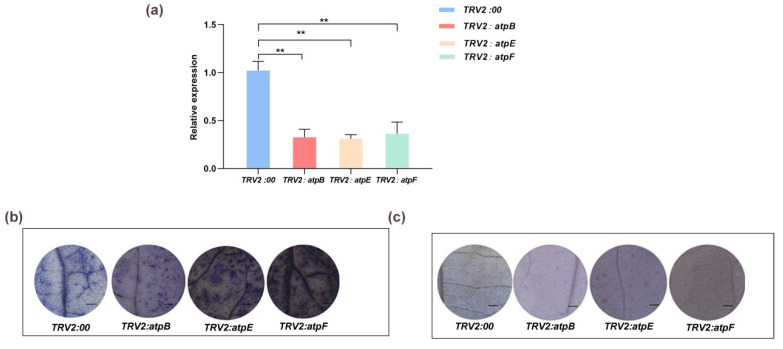
The NBT and DAB staining of gene-silenced cotton leaves. (**a**) Determination of gene expression of gene-silenced plants. Values are means ± SD of three replicates (** *p <* 0.01, Student’s *t*-tests); (**b**) NBT staining of gene-silenced plant leaves; (**c**) DAB staining of gene-silenced plant leaves. Bar = 1 mm.

**Figure 4 ijms-25-12707-f004:**
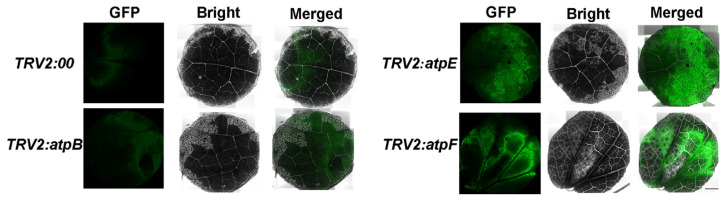
Determination of singlet oxygen in gene-silenced cotton leaves. SOSG detects ^1^O_2_ in leaves. GFP, green fluorescence; Bright, bright field; Merged, superposition field, bar = 1 mm.

**Table 1 ijms-25-12707-t001:** Chloroplast genome information of Jin A-CMS.

Encoding Gene	Genome Size (bp)	Gene Number	Gene Total Length (bp)	Gene Average Length (bp)	Gene Length/Genome (%)
	160,042	86	79,128	920	49.44
**Sample**	**Type**	**ncRNA Number**	**Total Length (bp)**	**Average Length (bp)**	**ncRNA Length/Genome (%)**
Jin A	tRNA	37	2826	76	1.77
Jin A	rrn23	2	5620	2810	3.51
Jin A	rrn4.5	2	206	103	0.13
Jin A	rrn16	2	2982	1491	1.86
Jin A	rrn5	2	242	121	0.15

**Table 2 ijms-25-12707-t002:** Chloroplast genome base composition information of Jin A-CMS.

Region	Length (bp)	T/U%	C%	A%	G%	AT%	GC%
Genome	160,042	31.83	19.04	30.86	18.27	62.69	37.31
LSC	88,623	33.17	18.18	31.54	17.11	64.71	35.29
IRb	25,599	28.50	20.68	28.48	22.34	56.98	43.02
SSC	20,221	34.43	16.60	33.85	15.13	68.28	31.72
IRa	25,599	28.48	22.34	28.49	20.69	56.97	43.03
Protein_coding_gene	79,128	31.18	17.84	30.56	20.42	61.74	38.26
First position	26,376	23.65	18.91	30.42	27.02	54.07	45.93
Second position	26,376	32.34	20.32	29.49	17.85	61.83	38.17
Third position	26,376	37.56	14.29	31.76	16.39	69.32	30.68
tRNA	2826	24.56	23.99	22.22	29.23	46.78	53.22
rRNA	9050	18.67	23.71	25.83	31.78	44.51	55.49

**Table 3 ijms-25-12707-t003:** Classification of gene function.

Category	Gene Group	Gene Name
Photosynthesis	Subunits_of_photosystem_I	*psaA psaB psaC psaI psaJ*
	Subunits_of_photosystem_II	*psbA psbB psbC psbD psbE psbF psbH psbI psbJ psbK psbL psbM psbN psbT psbZ*
	Subunits_of_NADH_dehydrogenase	*ndhA ndhB (×2) ndhC ndhD ndhE ndhF ndhG ndhH ndhI ndhJ ndhK*
	Subunits_of_cytochrome_b/f_complex	*petA petB petD petG petL petN*
	Subunits_of_ATP_synthase	*atpA atpB atpE atpF atpH atpI*
	Large_subunit_of_Rubisco	*rbcL*
Self-replication	Large_subunits_of_ribosome	*rpl2(×2) rpl14 rpl16 rpl20 rpl22 rpl23(×2) rpl32 rpl33 rpl36*
	Small_subunits_of_ribosome	*rps2 rps3 rps4 rps7(×2) rps8 rps11 rps12(×2) rps14 rps15 rps16 rps18 rps19*
	DNA-dependent_RNA_polymerase	*rpoA rpoB rpoC1 rpoC2*
	Ribosomal_RNAs	*rrn16 (×2) rrn23 (×2) rrn4.5 (×2) rrn5 (×2)*
	Transfer_RNAs	*trnA-UGC (×2) trnC-GCA trnD-GUC trnE-UUC trnF-GAA trnG-GCC trnG-UCC trnH-GUG trnI-GAU (×2) trnK-UUU trnL-CAA (×2) trnL-UAA trnL-UAG trnM-CAU (×4) trnN-GUU (×2) trnP-UGG trnQ-UUG trnR-ACG (×2) trnR-ACG trnR-UCU trnS-GCU trnS-GGA trnS-UGA trnT-GGU trnT-UGU trnV-GAC (×2) trnV-UAC trnW-CCA trnY-GUA*
Other genes	Maturase	*matK*
	Protease	*clpP1*
	Envelope_membrane_protein	*cemA*
	Acetyl-CoA_carboxylase	*accD*
	C-type_cytochrome_synthesis_gene	*ccsA*
Genes of unknown	Proteins_of_unknown_function	*ycf1 ycf2 (×2) ycf3 ycf4 ycf15 (×2)*

Note: *×2* present as a duplicate in the IR regions; *×4* present as three duplicates in the IR regions.

**Table 4 ijms-25-12707-t004:** Chloroplast protein-coding gene function annotation statistics of Jin A-CMS.

Sample ID	Total Protein	NR	GO	eggNOG	KEGG	Swiss-Prot
Jin A	86	84	67	54	73	86

**Table 5 ijms-25-12707-t005:** Chloroplast genome differential annotation information of Jin A-CMS.

Qeury Name	Qeury Length (bp)	Swiss_Tophit_Name	Swiss_Tophit_Description	Differential Amino Acid Location	Similarity (%)
*accD*	497	ACCD_GOSHI	Acetyl-coenzyme A carboxylase carboxyl transferase subunit beta	R-H (28), Y-S (126), P-S (136), Y-D (143), C-Y (159), S-Y (186)	98.8
*atpB*	498	ATPB_GOSHI	ATP synthase subunit beta	V-A (9), Q-R (52)	99.6
*atpE*	133	ATPE_GOSHI	ATP synthase epsilon chain	K-R (23)	99.2
*atpF*	189	ATPF_GOSHI	ATP synthase subunit b	S-G (50)	97.4
*ccsA*	320	CCSA_GOSHI	Cytochrome c biogenesis protein CcsA	I-L (78), S-A (183)	99.1
*clpP1*	196	CLPP_GOSHI	ATP-dependent Clp protease proteolytic subunit	T-K (279)	99
*ndhA*	363	NU1C_GOSHI	NAD(P)H-quinone oxidoreductase subunit 1	Y (24, −), R (122, +)	99.4
*ndhB*	510	NU2C2_GOSHI	NAD(P)H-quinone oxidoreductase subunit 2 B	G-A (57), A-V (288)	99.8
*ndhD*	506	NU4C_GOSHI	NAD(P)H-quinone oxidoreductase chain 4c	N-T (477)	99.6
*ndhF*	735	NU5C_GOSHI	NAD(P)H-quinone oxidoreductase subunit 5	L-F (48), I-L (479)	98.2
*ndhH*	393	NDHH_GOSHI	NAD(P)H-quinone oxidoreductase subunit H	D-Y (5), C-S (142), N-K (313), V-I (393), Q-K (395), S-N (422), R-S (489), T-P (498), N-K (503), R-L (517), V-L (623), L-F (628), L-F (648), V-F (721), F-L (724), F-L (726), S-Y (730)	99.7
*ndh* * I *	167	NDHI_GOSHI	NAD(P)H-quinone oxidoreductase subunit I	L-I (172)	99.4
*petA*	320	CYF_GOSHI	Cytochrome f	V-I (158)	99.4
*pet* *B*	215	CYB6_GOSHI	Cytochrome b6	A-V (176), T-A (211)	99.1
*psaA*	750	PSAA_GOSHI	Photosystem I P700 chlorophyll a apoprotein A1	I-V (143), E-D (156)	99.9
*psbK*	61	PSBK_GOSHI	Photosystem II reaction center protein K	S-A (370)	98.4
*rbcL*	479	PSBK_GOSHI	Ribulose bisphosphate carboxylase large chain	F-L (48)	99
*rpl16*	135	RK16_GOSBA	50S ribosomal protein L16	E-Q (28), D-H (86), F-Y (226), M-V (255)	99.3
*rpl22*	151	RK22_GOSBAcc	50S ribosomal protein L22	N-T (7)	99.3
*rpoA*	329	RPOA_GOSBA	DNA-directed RNA polymerase subunit alpha	M-R (8), T-A (70), I-T (142)	99.1
*rpo* *B*	1070	RPOB_GOSHI	DNA-directed RNA polymerase subunit beta	T-A (234), Y-S (273), I-M (311)	99.9
*rpo* *C1*	689	RPOC1_GOSHI	DNA-directed RNA polymerase subunit beta	P-T (302)	99.1
*rpo* *C2*	1393	RPOC2_GOSHI	DNA-directed RNA polymerase subunit beta	V (145, +), Y (146, +), P (147, +), N (148, +), Q-R (607), S-L (627)	99
*rps15*	90	RR15_GOSHI	30S ribosomal protein S15	R-L (384), S-N (499), E-D (535), M-I (599), P-Q (694), Q-R (801), A-S (944), F-L (1010), H-D (1057), T-I (1125)	98.9
*rps1* *6*	88	RR16_GOSHI	30S ribosomal protein S16	A-V (22)	98.9
*rps* *3*	218	RR3_GOSHI	30S ribosomal protein S3	T-S (81), T-N (88)	99.5
*rps* *8*	134	RR8_GOSBA	30S ribosomal protein S8	T-M (51)	99.3
*ycf1*	1893	TI214_GOSHI	Protein TIC 214	S-Y (132)	97.9
*ycf* *2*	2298	YCF2_GOSHI	Protein Ycf2	L-F (3), F-V (49), N-K (165), P-T (255), S-Y (291), M (303,+), D (304, +), I-R (319), G-D (321), F-L (323), R-G (326), Y-S (332), W-L (443), D-E (553),C-R (757), S-D (765), I-L (992),D-E(1071), H-D (1317), H-N (1364), D-H (1446), K-N (1466), I-L (1478), Q-K (1561), F-L (1609), D-G (1841)	99.5

Values are means ± SD of three replicates (* *p <* 0.05; ** *p <* 0.01, Tukey’s multiple comparison tests).

## Data Availability

The original contributions presented in the study are publicly available. These data can be found at https://www.ncbi.nlm.nih.gov/bioproject/PRJNA1106185.

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
