# Peer review of "Effect of Chloroplast ATP Synthase on Reactive Oxygen Species Metabolism in Cotton"

_ijms, 2024, doi:10.3390/ijms252312707_

Round 1
Reviewer 1 Report
Comments and Suggestions for Authors
Dear Authors,
Reviewer comments ijms-3325090
The manuscript entitled „Effect of chloroplast ATP synthase on reactive oxygen species metabolism in cotton“ represents a useful study aimed at an investigation of chloroplast genome in cotton with a focus on ATP synthase genes expresion and their relationship to ROS accumulation, namely the fact that atpE and atpF gene silencing led to an enhanced ROS accumulation in cotton leaves.
I can recommend the manuscript for publication in IJMS.
However, I have some minor comments on the present manuscript:
1/ The finding that atpE and atpF gene silencing leads to an enhanced ROS accumulation in cotton leaves represents a very interesting new result. Do the authors have some hypothesis on the relationship between ATP synthase components and ROS levels regulation in plant chloroplasts? I think that some hypothesis has to be presented in Discussion.
2/ In Materials and methods, when public databases are cited a date of access has to eb added to each public database cited since the content available in public databases can vary with time.
3/ In Materials and methods, Statistical analysis, the authors wrote that they used Tukey´s multiple comparisons test for the data evaluation. However, from the presented figures, it seems that only pairs of samples were compared in Figure 2 and Figure 3a thus T-test should be used for a comparison of a pair of samples.
4/ Formal comments on text related to English language and style:
Introduction, line 48: Modify the word form „still existent“ to „still exist“.
Lines 300,302: Replace the word „less“ with „lower“ in the statement „reads lower than 200 bp in length…“
Line 316. Modify the word form „partitions“ to „parts“ in the statement „…and the 4 chloroplast parts were determined.“
Line 383: A reference on GraphPad Prism 8 software has to be added.
Final recommendation: Accept after a minor revision.
Author Response
|
1. Summary |
|
|
|
Thank you very much for taking the time to review this manuscript. Please find the detailed responses below and the corresponding revisions/corrections highlighted/in track changes in the re-submitted files. |
||
2. Point-by-point response to Comments and Suggestions for Authors |
|
|
|
Comments 1: The finding that atpE and atpF gene silencing leads to an enhanced ROS accumulation in cotton leaves represents a very interesting new result. Do the authors have some hypothesis on the relationship between ATP synthase components and ROS levels regulation in plant chloroplasts? I think that some hypothesis has to be presented in Discussion. Response 1: Thank you for pointing this out. I agree with this comment. Therefore, I have speculated on the relationship between ATP synthase components and ROS levels regulation by referring to the existing literature in discussion. Comments 2: In Materials and methods, when public databases are cited a date of access has to eb added to each public database cited since the content available in public databases can vary with time. Response 2: I agree with this comment. Therefore, I have added the date in methods. Comments 3: In Materials and methods, Statistical analysis, the authors wrote that they used Tukey´s multiple comparisons test for the data evaluation. However, from the presented figures, it seems that only pairs of samples were compared in Figure 2 and Figure 3a thus T-test should be used for a comparison of a pair of samples. Response 3: I agree with this comment. Therefore, I have made some corrections to the notes. Comments 4: Formal comments on text related to English language and style: Introduction, line 48: Modify the word form „still existent“ to „still exist“. Lines 300,302: Replace the word „less“ with „lower“ in the statement „reads lower than 200 bp in length…“ Line 316. Modify the word form „partitions“ to „parts“ in the statement „…and the 4 chloroplast parts were determined.“ Line 383: A reference on GraphPad Prism 8 software has to be added. Response 4: I agree with this comment. Therefore, I have added it. And I have also revised and checked the English language of the manuscript. |

Reviewer 2 Report
Comments and Suggestions for Authors
This manuscript presents an interesting study on the relationship between chloroplast genes, particularly ATP synthase subunits genes (atpE and atpF), and reactive oxygen species (ROS) metabolism in cytoplasmic male sterility (CMS) in cotton. These results provide a basic information for the analysis of ATP synthase functions in cotton. However, there are several areas that need improvement before publication.
1.In the “Introduction” section, the connection between the abnormal programmed cell death in the tapetum induced by ROS and the study’s focus on chloroplast genes in Jin A - CMS could be more clearly stated at the beginning. It seems a bit of a jump from the general introduction of CMS caused by ROS and genetic interactions to specifically looking at chloroplast genes in Jin A without a more explicit link.
2.In the “Introduction” section, the role of ROS in the induction of PCD can be described in more detail, such as “reactive oxygen species (ROS) are the primary inducers of PCD in plants. Excessive ROS accumulation can lead to oxidative stress, which triggers the PCD pathway and ultimately results in cell death.”
3.In addition to quality control of the genome sequencing data, the authors also need to analyze the frequency distribution of bases in the genome data to check whether there is any base separation of A, T and G, C. In addition, the authors need to analyze the base quality of the sequencing data. Based on the above three assessments of the quality of the sequencing data, it can be shown that the data produced by the sequencing are of normal quality with a low error rate, and can be used for follow-up work.
4.In the results section, when comparing single nucleotide polymorphisms (SNPs) in the chloroplast genes of Jin A - CMS and land cotton (Gossypium hirsutum), in addition to listing the differential genes and their amino acid changes, the potential impact of these SNPs on the gene function can be further analyzed, e.g., by using bioinformatic prediction tools to analyze whether the amino acid changes are located in functionally critical regions, thereby explain in more depth the relationship of these gene differences with CMS and ROS metabolism.
5.In the discussion section, when a possible mechanism of action of chloroplast ATP synthase in CMS is proposed, it can be combined with the existing literature to discuss how chloroplast ATP synthase affects programmed cell death (PCD) by regulating energy metabolism and ROS levels.
6.In the discussion section, it is suggested to compare the results of the present study with similar studies already available, for example, by mentioning the role of chloroplast ATP synthase in ROS metabolism in other plants.
7.The text contains several grammatical errors and inaccuracies in presentation that require careful proofreading and revision.
8.Minor comments:
(1)Lines 197 - 203 do not belong to the content of the results. They belong to the section of Materials and Methods.
(2)Table 5 needs to be supplemented with units for Qeury length and Similarity.
(3)In figure 3, “P < 0.05” should be replaced with “P < 0.05”. Please check the full manuscript and make corrections.
(4)In 4.5 part, it is recommended to supplement the reaction conditions for the ligand gene CDS and the plasmid pMD19T.
Comments on the Quality of English LanguageThe English language should be improved.
Author Response
|
1. Summary |
|
|
|
Thank you very much for taking the time to review this manuscript. Please find the detailed responses below and the corresponding revisions/corrections highlighted/in track changes in the re-submitted files. |
||
2. Point-by-point response to Comments and Suggestions for Authors |
|
|
|
Comments 1: In the “Introduction” section, the connection between the abnormal programmed cell death in the tapetum induced by ROS and the study’s focus on chloroplast genes in Jin A - CMS could be more clearly stated at the beginning. It seems a bit of a jump from the general introduction of CMS caused by ROS and genetic interactions to specifically looking at chloroplast genes in Jin A without a more explicit link. Response 1: Thank you for pointing this out. I agree with this comment. Therefore, I have revised and combed the introduction to make it more coherent. Comments 2: In the “Introduction” section, the role of ROS in the induction of PCD can be described in more detail, such as “reactive oxygen species (ROS) are the primary inducers of PCD in plants. Excessive ROS accumulation can lead to oxidative stress, which triggers the PCD pathway and ultimately results in cell death.” Response 2: I agree with this comment. Therefore, I have added it. Comments 3: In addition to quality control of the genome sequencing data, the authors also need to analyze the frequency distribution of bases in the genome data to check whether there is any base separation of A, T and G, C. In addition, the authors need to analyze the base quality of the sequencing data. Based on the above three assessments of the quality of the sequencing data, it can be shown that the data produced by the sequencing are of normal quality with a low error rate, and can be used for follow-up work. Response 3: I agree with this comment. Therefore, I have added the methods. Comments 4: In the results section, when comparing single nucleotide polymorphisms (SNPs) in the chloroplast genes of Jin A - CMS and land cotton (Gossypium hirsutum), in addition to listing the differential genes and their amino acid changes, the potential impact of these SNPs on the gene function can be further analyzed, e.g., by using bioinformatic prediction tools to analyze whether the amino acid changes are located in functionally critical regions, thereby explain in more depth the relationship of these gene differences with CMS and ROS metabolism. Response 4: I agree with this comment. In future studies, we will focus on exploring more functions of these differential genes through bioinformatics methods and functional validation. |
|
|
|
Comments 5: In the discussion section, when a possible mechanism of action of chloroplast ATP synthase in CMS is proposed, it can be combined with the existing literature to discuss how chloroplast ATP synthase affects programmed cell death (PCD) by regulating energy metabolism and ROS levels. Response 5: Thank you for pointing this out. I agree with this comment. Therefore, I have speculated on the relationship between ATP synthase components and ROS levels regulation by referring to the existing literature in discussion.
Comments 6: In the discussion section, it is suggested to compare the results of the present study with similar studies already available, for example, by mentioning the role of chloroplast ATP synthase in ROS metabolism in other plants. Response 6: Thank you for pointing this out. I agree with this comment. Therefore, I have added it in Discussion. Comments 7: The text contains several grammatical errors and inaccuracies in presentation that require careful proofreading and revision. Response 7: Thank you for pointing this out. I agree with this comment. Therefore, I have made a correction. Comments 8: Minor comments: (1)Lines 197 - 203 do not belong to the content of the results. They belong to the section of Materials and Methods. (2)Table 5 needs to be supplemented with units for Qeury length and Similarity. (3)In figure 3, “P < 0.05” should be replaced with “P < 0.05”. Please check the full manuscript and make corrections. (4)In 4.5 part, it is recommended to supplement the reaction conditions for the ligand gene CDS and the plasmid pMD19T. Comments on the Quality of English Language The English language should be improved. |
|
Response 8: I agree with this comment. Therefore, Therefore, I have added it and revised and checked the English language of the manuscript.

Round 2
Reviewer 2 Report
Comments and Suggestions for Authors
Thank you for your revision.
There are some minor suggestions:
(1) line 56: maintains->maintain
(2) line 338: The phrase 'accessed on 9 November 2023' should be placed in parentheses.
Author Response
comment 1: line 56: maintains->maintain
response 1: Thank you for your valuable comments, I have revised.
comment 2: line 338: The phrase 'accessed on 9 November 2023' should be placed in parentheses.
response 2: Thank you for your valuable comments, I have revised.
